# Semantic Conflict Resolution for Access and Usage Control

Eugenia I. Papagiannakopoulou[1,*,†], Nikolaos L. Dellas[1,*,†], Georgios V. Lioudakis[1,*,†], Maria N. Koukovini[1,*,†] and Aziz Mousas[1,*,†]

[1]*ICT abovo P.C., 20 Iridanou St, Athens, 11528, Greece*

### Abstract

Considering the use case when organisations acquire data from third parties and must align their processing to access and usage constraints set forth by the data providers, this paper targets a twofold goal: to allow for semantic interoperability of policies defined by different parties, and to allow for semantic conflict identification between these policies, and provide the means for their comprehensive resolution. In this context, the policy engine of goodFlows, a semantic access and usage control framework, has been adapted in order to incorporate the mechanisms for the semantic alignment of its information model with standardised ontologies, as well as for effectively identifying and resolving conflicts between heterogeneous policies. To this end, the main results of this paper are: an ontology-import mechanism that automatically aligns organisational information models with standard vocabularies such as DPV and ODRL; a bi-directional ODRL adaptor that translates ODRL structures into semantic attribute-based access and usage control rules and vice versa; and an enhanced Policy Decision Point (PDP) that applies jurisprudential precedence principles and is able to identify conflicts in complex settings and propose appropriate resolutions.

### Keywords

Access and usage control, W3C ODRL, W3C DPV, ontology alignment, policy conflict resolution

## 1. Introduction

The European Strategy for Data [1] sets the goal of a single market for data built around Common European Data Spaces, so that more data can circulate while the original holders retain control. However, as also stressed by both the Data Governance Act (DGA) [2] and the Data Act [3], today's landscape is fragmented; the reasons for this fragmentation include implications as regards semantic and technical interoperability, as well as the reluctance of data holders in sharing their data due to the potential of losing control regarding their use. The latter is greatly an issue of interoperation; when a potential data consumer tries to ingest a third-party dataset, it must adhere to the provider's access and usage constraints and the lack of automation thereof often undermines synergies.

In light of these issues, this paper has a twofold goal: On the one hand, to allow for semantic interoperability of policies defined by different parties. On the other hand, to allow for semantic conflict identification between these policies, and provide the means for their comprehensive resolution.

### 1.1. Related Work

Existing policy languages such as the eXtensible Access Control Markup Language (XACML) 3.0 [4] and the W3C Open Digital Rights Language (ODRL) 2.2 [5][6] provide syntactic interoperability but only basic conflict-handling. XACML relies on combining algorithms (e.g., deny-overrides, permit-overrides, first-applicable) to choose a single outcome if rules disagree. Similarly, the built-in conflict resolution strategy of ODRL relies on a static conflict property (perm, prohibit, or invalid) and is insufficient for handling complex rule conflicts in dynamic environments [7]. These limitations are inherited by

*NeXt-generation Data Governance workshop 2025 (NXDG 2025), co-located with SEMANTiCS'25: International Conference on Semantic Systems, September 3–5, 2025, Vienna, Austria*

*Corresponding author.

†These authors contributed equally.

✉ eugenia.papagiannakopoulou@ict-abovo.gr (E. I. Papagiannakopoulou); nikolaos.dellas@ict-abovo.gr (N. L. Dellas); georgios.lioudakis@ict-abovo.gr (G. V. Lioudakis); mariza.koukovini@ict-abovo.gr (M. N. Koukovini); azmousas@ict-abovo.gr (A. Mousas)

approaches leveraging ODRL for conflict resolution, such as the Open Digital Rights Enforcement framework (ODRE) [8], or the one proposed by Gaia-X [9]. A recent survey [10] has reviewed more than twenty policy languages and confirms that only a handful support semantic conflict resolution, i.e., the ability to explain or repair clashes rather than merely flag them. Fostering to allow for negotiation in consent banners (cookies) on the Web, the work in [11] builds upon ODRL and introduces a Compliance Report Model to denote the result of an ODRL evaluation in an interoperable manner and test suite for letting different evaluators agree on the result of a policy check, but their engine still flags conflicts rather than merging or rewriting rules. Recent work on the challenges of policy consistency checking [12] throws light on logic-driven approaches such as the framework described in [13]; however, the presented DSA-Analyser outputs either the confirmation that no conflicts arise among the evaluated rules or the complete list of conflicts, each of them associated to the related context, with no mechanisms for conflict resolution.

Several initiatives have started to align ontologies related to usage control. The W3C DPVCG, that develops and maintains the Data Privacy Vocabulary (DPV) [14], recently published a draft mapping between DPV and ODRL [15], so that privacy concepts can be reused inside ODRL policies. The work in [16] extends the Data Use Ontology (DUO) [17] for genomic data by encoding ODRL rules and DPV annotations, demonstrating better matchmaking, also comprehensively dealing with conflict resolution, based on the work conceived in [18], without however considering the *lex posterior* principle, which is an innovative aspect of our work.

The paper advances the state-of-the-art in policy-based access and usage control, by providing innovative contributions focusing on interoperability and collaboration. It provides the means for aligning proprietary models with standardised ontologies such as ODRL and DPV, and bridges the gap between ODRL and attribute-based access control, thereby allowing for comprehensive resolution of conflicts arising between policies specified by different parties leveraging advanced conflict resolution strategies beyond the static approach of ODRL.

## 1.2. Context and contribution of this paper

In the data space scenarios addressed by this paper, there are two roles that interact:

- A data provider, such as a data holder ([3]), that publishes a dataset together with an ODRL policy stating the permitted and forbidden actions, the conditions under which they apply, and any additional obligations on the consumer.
- A data consumer, that designs a Data Processing Workflow (DPW) that includes processing of the provider's dataset, thereby requiring adherence to the constraints set forth by the provider.

In this context, interoperability issues may emerge, due to the use of different semantic models, as well as conflicts between consumer's processing intentions and provider's constraints regarding the use of data. Such conflicts may not only concern contradictory rules, but also variations of similar authorisation contexts, such as different data encryption status, or contextual attributes; to this end, rules correlation should not concern their syntactic comparison, but rather their semantic analysis.

Therefore, the focus of this paper is to foster interoperability and collaboration among data providers and consumers, by providing three mechanisms:

- An ontology-import mechanism that automatically aligns organisational information models with standard vocabularies such as DPV and ODRL.
- A bi-directional ODRL adaptor that translates ODRL structures into semantic attribute-based access and usage control rules and vice versa.
- An enhanced Policy Decision Point (PDP) that is able to identify conflicts in complex settings and propose appropriate resolutions.

These mechanisms have been incorporated to goodFlows, a semantic framework devised for the specification of access and usage policies and workflows over a common information model, and the

verification and re-engineering of workflows on the basis of the policies. The results presented in this paper have been mainly developed in the context of the Horizon Europe UPCAST project[1].

The rest of the paper is outlined as follows. Section 2 provides some background information on goodFlows. In Section 3, the two alignment mechanisms are described, fostering interoperability at the information and policy model levels. Section 4 delves into the process of resolving conflicts between data consumer's processing intentions and data provider's constraints. Finally, Section 5 provides some concluding remarks.

## 2. Background: goodFlows

goodFlows is a product of several years of research and development [19][20], that has matured in the context of the H2020 BPR4GDPR project[2] [21]. It has been conceived as a holistic framework designed to aid organisation of various sizes in ensuring compliance with the General Data Protection Regulation (GDPR) [22]; it ensures that processes, and workflows in general, are automatically aligned with both organisational goals and GDPR requirements, by design and by default. In more detail, goodFlows enables organisations to formally define their information model in sufficient detail, with a strong emphasis on concepts and assets relevant to GDPR. On the basis of this information model, it allows for the definition of policies containing any applicable GDPR-related rules as adapted to the specific needs of the organisation, in a user-friendly, flexible, and fine-grained way. Processes and workflows can be modelled using an easy-to-use no-code design and editing tool, which supports automatic verification of whether a workflow model is GDPR-compliant; where possible, non-compliant workflow models can be automatically transformed into compliant ones.

The framework is steered by two main pillars: a rule-based access and usage control framework, and a system for compliant workflow planning and re-engineering (Figure 1). The former comprises a Policy Decision Point (PDP) that, based on the afore-mentioned organisational information model, performs all reasoning required for taking authorisation decisions and driving the workflow verification and re-engineering towards compliance performed by the latter.

Underpinning this, there are three foundational models, implemented as semantic ontologies (Figure 2): the information model, the policy model, and the workflow model. Reasoning in the policy model results in the formation of *Compliance Directives*, being ontological structures used to regulate the verification and transformation procedure [23]. Together, these elements form an integrated environment that facilitates the development and maintenance of compliant, privacy-aware business processes and workflows.

However, goodFlows was characterised by two limitations. On the one hand, there was no means to import and align standard ontologies. On the other hand, whereas goodFlows' PDP allowed for complex reasoning, it was lacking the mechanisms to import and take into consideration the constraints of data subjects and data holders, thereby resolving conflicts between these constraints and internal policies and workflows. These two limitations have motivated this work, and the results towards their resolution are presented in this paper.

### 2.1. Information Model

Both policies and processes are defined on the basis of the organisation's Information Model. To this end, the goodFlows Information Model Editor, a user-friendly software application devised for the specification and management of the underlying semantic model, provides all necessary functionality for the management of all different information classes and their relations, and the corresponding Information Model Ontology (IMO). The use of the editor hides the technical details of the model, requiring no particular technical expertise by its users; it translates the input provided through the graphical interface to machine code.

---

[1]https://www.upcast-project.eu/
[2]https://www.bpr4gdpr.eu/

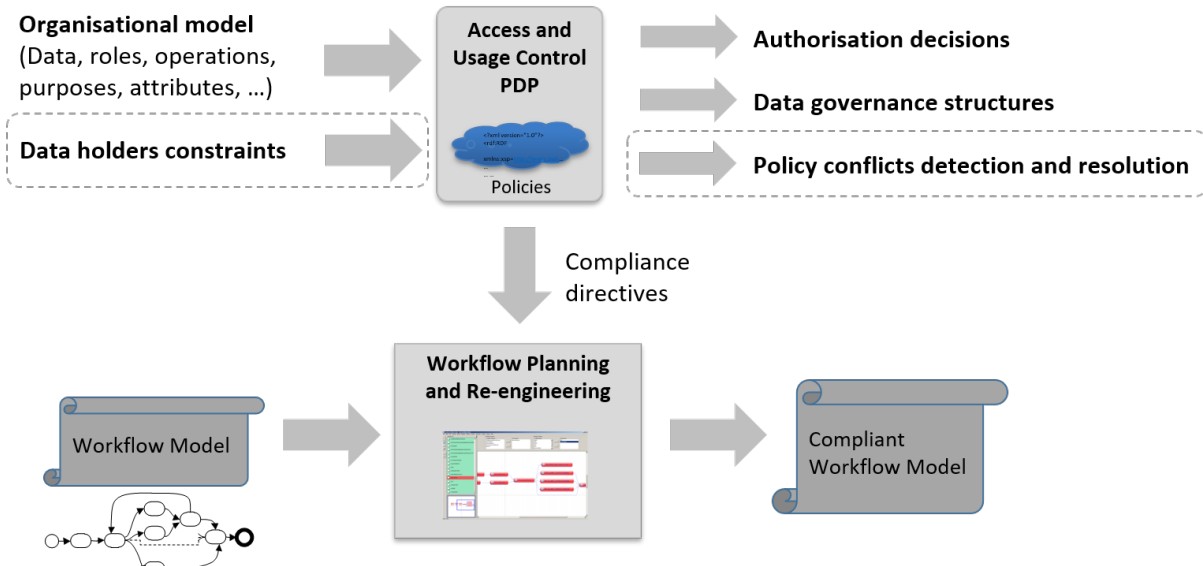

**Figure 1:** goodFlows concept (dashed rectangles highlight contribution of this work)

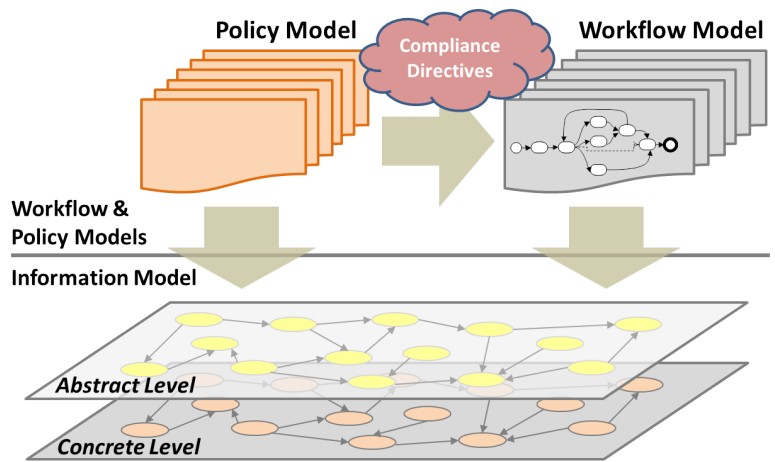

**Figure 2:** goodFlows models and ontologies

A consumer's domain-specific information model is specified through the respective editor, modelling data types, purposes, operations, roles, etc., met in the operation of the organisation. In that respect, the Information Model ontology includes all associated classes, e.g., `DataTypes`, `Purposes`, `Roles`, `Operations` and `ContextTypes`, while individuals of each class comprise AND- and OR- hierarchies. Figure 3 illustrates the `DataTypes`' graph of a model. As shown, entities in the graph are interconnected with two types of relations; blue lines denote `isA` specialisation relation, whereas red lines reflect the `isPartOf` inclusion relation.

The formed hierarchies allow the definition of rules upon high-level entities which will be propagated across the hierarchies, alleviating the need to explicitly define exhaustive rulesets. In that respect, both positive and negative authorisations will be propagated to more specific entities, while negative authorisations will also be propagated across `isPartOf` relations.

## 2.2. Policy Model

In goodFlows, an access and usage control rule is specified through the following structure:

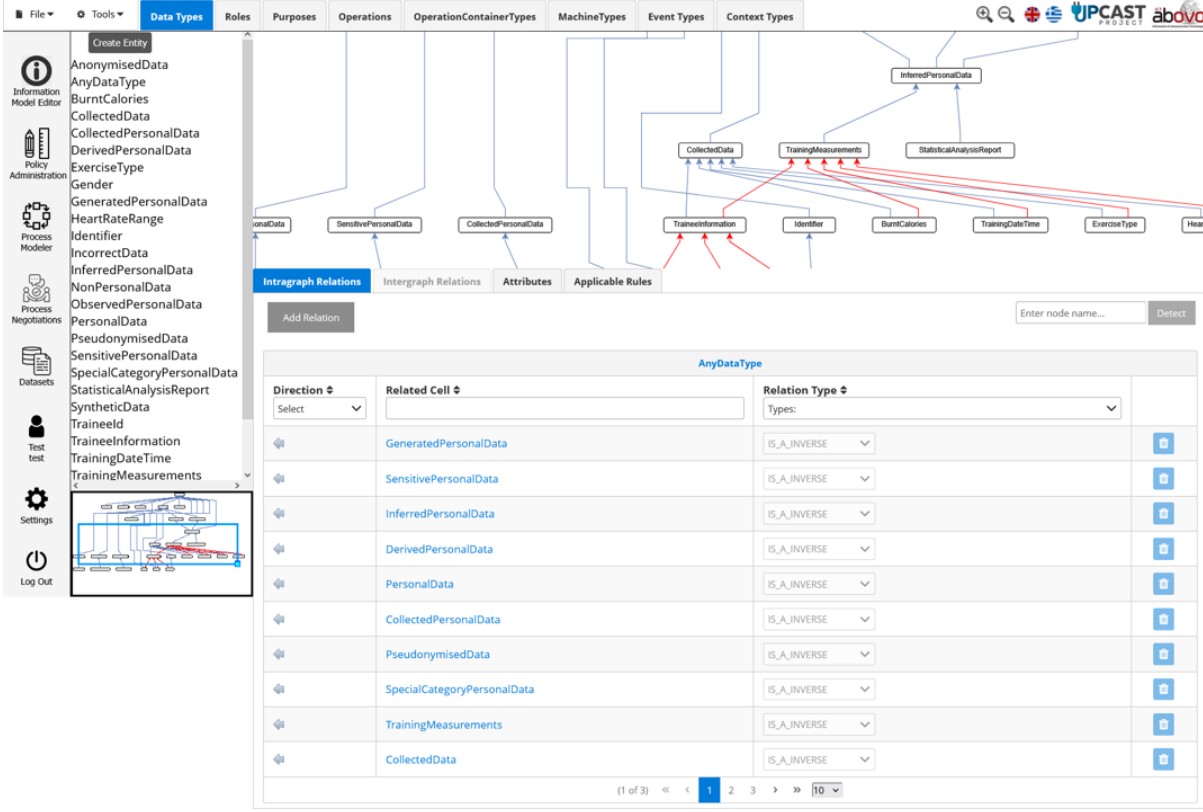

**Figure 3:** Hierarchies within the `DataTypes`' graph

$$\left.\begin{array}{l} \textit{Permission} \\ \textit{Prohibition} \\ \textit{Obligation} \end{array}\right\} (\textit{pu, act, preAct, cont, postAct})$$

where *act* is the action that the rule applies to; *pu* is the purpose for which *act* is permitted/prohibited/obliged to be executed; *cont* is a structure of contextual parameters; *preAct* is a structure of actions that should have preceded; *postAct* refers to the action(s) that must be executed following the rule enforcement.

Actions constitute conceptual triples of *actor, operation, resource* and are formed leveraging the entities of the Information Model. In more detail, action's entities are defined as a tuple ⟨*concept, constraint*⟩, where *concept* is an Information Model entity, and *constraint* is an expression (or a logical relation thereof) assigning values to attributes and/or sub-concepts of the given entity. This mechanism allows the definition of fine-grained access and usage control rules inline with the Attribute-based Access Control (ABAC) paradigm.

Towards improved performance during real-time operation, such as the validation of processes against policies, offline reasoning, i.e., proactive extraction of knowledge contained in the access and usage control rules, is leveraged. As the Policy Model considers various hierarchies, rules defined at a high level of abstraction are propagated across the corresponding IMO graphs. Thus, starting from the explicitly specified rules, initially comprising the Rules (RU) set, meta-rules are generated, following specific inheritance patterns. The latter are based on the Information Model specialisation (`isA`) and inclusion (`isPartOf`) properties, while they depend on the type of the rule, that is whether it is a permission, a prohibition, or an obligation.

The Policy Administration editor of goodFlows (Figure 4) provides functionality for rule management, as well as extracting meta-rules from explicitly defined ones for high-level entities, while it guides the user through the definition of a rule's elements in a user-friendly manner. Essentially, it provides Policy Administration Point (PAP) and Policy Decision Point (PDP) functionalities; it allows the specification

of fine-grained access and usage policies and it plays a critical role in compliance validation by checking workflows against organisational policies and third-party usage constraints. For instance, Figure 4 depicts the creation of a rule stating that an organisation of type `NonGovernmentalOrganisation` may make available (`MakeAvailable`) inferred personal data (`InferredPersonalData`) if these data are aggregated (`Status Equals Aggregated`) and within the EU (contextual attribute `spatial` Equals `EU`), for the purpose of `ResearchAndDevelopment`.

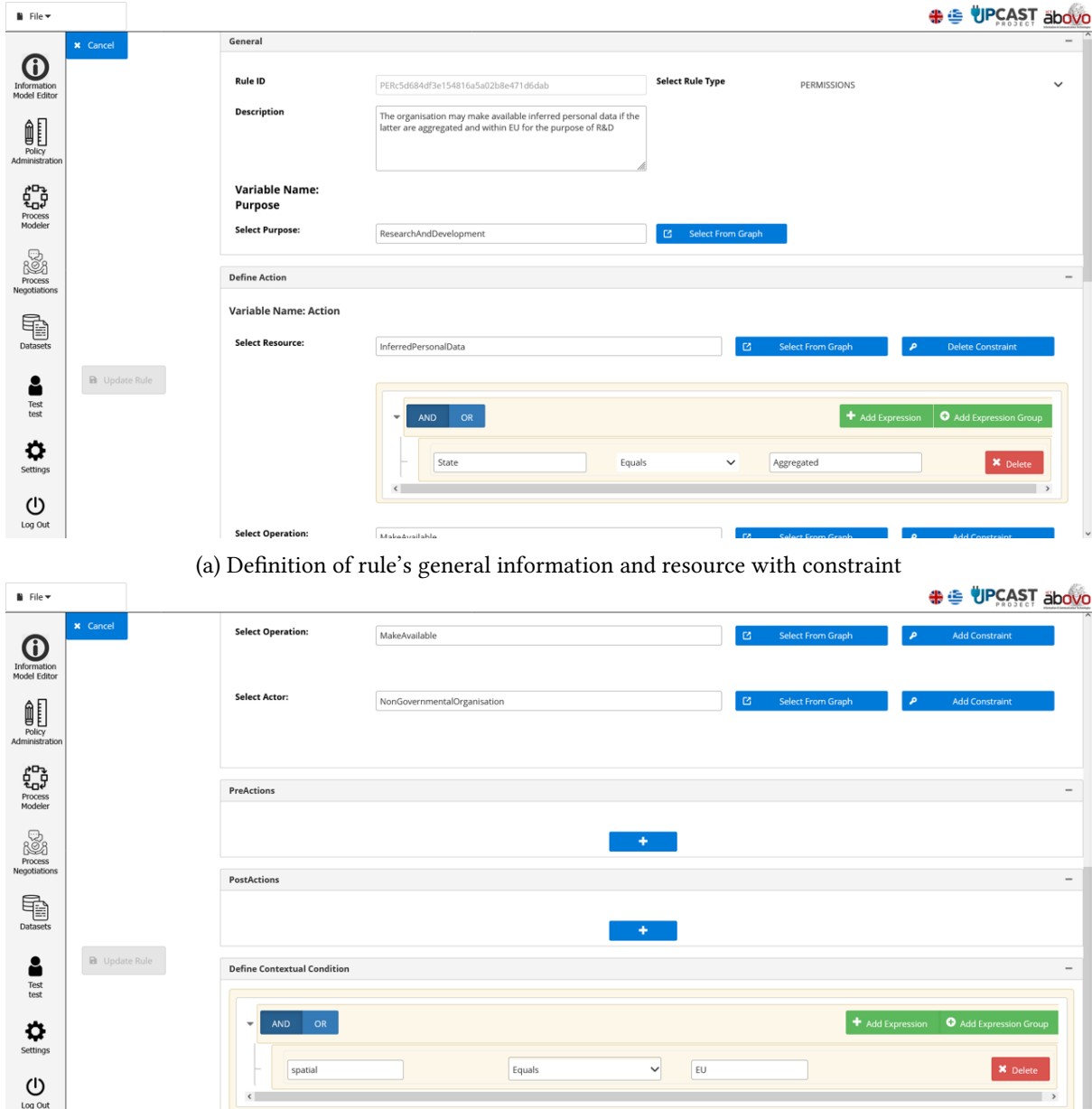

(a) Definition of rule's general information and resource with constraint

(b) Definition of operation, actor, contextual constraints, and pre-/post-actions

**Figure 4:** Rule creation in goodFlows

## 2.3. Workflow Model

The goodFlows workflow modelling approach has been designed with the aim to inherently support the prescription of provisions necessary for GDPR compliance and as such comprises the compliance metamodel, implemented as an ontology. The most fundamental artefacts of a workflow model are *tasks* and *flows*. The former represent actions to be executed within the workflow, each describing

the operation performed by an actor on an asset under specific conditions. Tasks in goodFlows can be seen as the cumulative effect of certain actors performing operations on well-defined assets. In order to allow for more flexibility in the definition of workflows, the tool has introduced the concept of Execution Profiles for defining distinct ⟨*actor*, *operation*, *asset*⟩ combinations as needed, which may therefore be used to denote alternative modes of executing the same task. Flows express control and data dependencies between tasks and are represented through directed edges. Further, a workflow model is complemented by the operational *purposes* it is meant to serve, and the potential *initiators*, denoting entities authorised to initiate the workflow.

In essence, workflow specification via the dedicated goodFlows Process Modeller environment constitutes the primary means through which the data consumer states their intentions for the data they seek to acquire or that they have acquired. These intentions are derived from jointly considering a variety of aspects, including: the processing operations intended to be performed; the entities in direct or indirect control of their execution; the attributes of the acquired data and the conditions under which any processing and exchange is meant to take place; the stated purposes that the process in question is intended to serve.

Figure 5 illustrates the Process Modeller functionality, modelling a simple Data Processing Workflow (DPW) that will drive the examples provided in the paper. A dataset of type `TrainingMeasurements` is fed to a task performing statistical analysis, and the resulting statistical analysis report is subsequently published; intended purpose for the execution of the DPW as a whole is defined to be `CommercialResearch`.

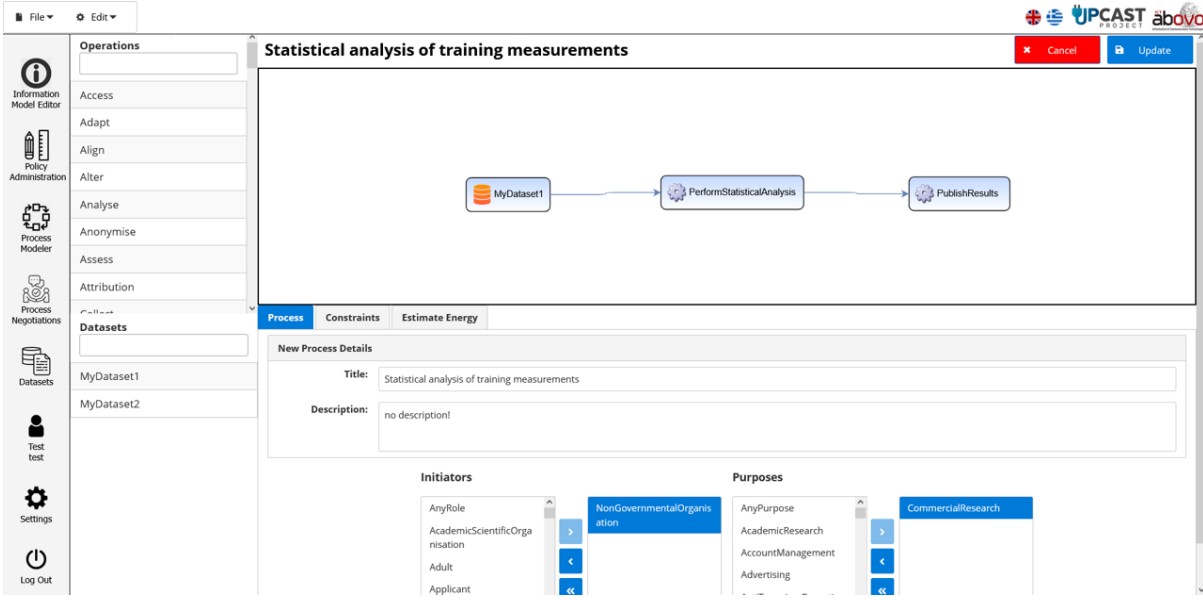

**Figure 5:** Process Modeller

## 3. Alignment with ODRL and DPV

This section outlines the two alignment mechanisms proposed, fostering interoperability at the information and policy model levels.

### 3.1. Information Model alignment

In order to foster semantic interoperability and compliance to standards, appropriate functionality for importing third-party ontologies and aligning them with the goodFlows Information Model ontology has been implemented. The requirements for a generic ontologies' import functionality were mainly derived from the structure and alignment needs of the DPV and ODRL ontologies; class mappings to

**Table 1**
goodFlows IM – DPV/ODRL mapping

| goodFlows IM | DPV | ODRL | Comments |
|---|---|---|---|
| Class individuals | Classes | Class individuals | Both goodFlows IM and the ODRL ontology represent vocabulary for a class as class individuals, whereas DPV defines them as subclasses |
| IM:Purposes class | dpv:Purpose class | - | dpv:Purpose class corresponds to IM:AnyPurpose instance. All subclasses of dpv:Purpose are inserted as IM:Purposes class individuals |
| IM:Roles class | dpv:Entity class | - | dpv:Entity class corresponds to IM:AnyRole instance. All subclasses of dpv:Entity are inserted as IM:Roles class individuals |
| IM:Operations class | dpv:Processing | odrl:Action | dpv:Processing class corresponds to IM:AnyOperation instance. All subclasses of dpv:Processing are inserted as IM:Operations class individuals |
| IM:DataTypes class | dpv:Data | - | dpv:Data class corresponds to IM:AnyDataType instance. All subclasses of dpv:Data are inserted as IM:DataTypes class individuals |
| IM:ContextTypes class | dpv:Context, dpv:Location | odrl:LeftOperand | dpv:Context class corresponds to IM:AnyContextType instance. All subclasses of dpv:Context and dpv:Location are inserted as IM:ContextTypes class individuals |
| IM:Attributes class | ObjectProperty, DatatypeProperty | ObjectProperty, DatatypeProperty | IM:Attributes individuals are generated for properties defined in ODRL and DPV ontologies. |
| IM:isA | subclassing | odrl:includedIn | DPV subclassing forms specialisation hierarchies in IM leveraging IM:isA relationship. Specialisation relations between ODRL individuals are translated to IM:isA |

the equivalent goodFlows Information Model classes, as well as handling of hierarchical and other relationships, provided key insights into designing a flexible import mechanism that supports subclassing, specialisation and inclusion properties, and multi-step ontology integration. Table 1 summarises the findings regarding alignment of goodFlows Information Model and DPV and ODRL.

Based on the findings of Table 1, that can be generalised for importing and aligning any third-party ontology, a dedicated structure for the import configuration has been designed. It follows a structured approach that allows, e.g., for both subclassing and class individual mappings, depending on the ontology being imported. The configuration is represented as a JSON array of mappings, where each object corresponds to a different ontology import; each mapping object contains the following key elements:

- `classesMappings`: Defines how concepts from the imported ontology are mapped to goodFlows IM classes.
    - `sourceClasses`: A list of one or more ontology classes that are mapped to a single goodFlows IM class. For example, both "Context" and "Location" from DPV are mapped to "ContextTypes", ensuring semantic consistency.
    - `targetClass`: Specifies the corresponding goodFlows IM class where the source classes should be aligned.

- `namespace` and `namespaceDelimiter`: Define the imported ontology's namespace and its delimiter, ensuring that imported entities are correctly referenced.
- `orderOfImporting`: Specifies the sequence in which ontologies are processed, thus ensuring dependencies are resolved correctly.
- `subclassing`: Determines whether the imported ontology forms the vocabulary associated with a class as subclasses (true), as in DPV case, or as class individuals (false), as in ODRL case.
- `specialisationProperties`: Complementary to subclassing processing, in case the last is set to false. Present in the import configuration, it declares specialisation relationships between imported ontology individuals (e.g., odrl:includedIn and skos:broaderTransitive used in ODRL) to be translated into `isA` relations in goodFlows IM.
- `inclusionProperties`: Present in the import configuration, this property declares inclusion relationships between imported ontology individuals to be translated into `isPartOf` relations in goodFlows IM.

Figure 6 shows an excerpt from the goodFlows Information Model ontology created after importing and aligning the DPV and ODRL ontologies; the `Derive` individual in the Information Model ontology has been aligned with equivalent concepts in both the DPV and ODRL ontologies. As part of the ontology import and alignment process, two owl:sameAs properties have been inserted, associating "http://www.ict-abovo.eu/ontologies/InformationModel#Derive" to "https://w3id.org/dpv/dpv-owl#Derive" from DPV and "http://www.w3.org/ns/odrl/2/derive" from ODRL. This ensures semantic interoperability by explicitly stating that these concepts represent the same meaning across different ontologies.

```
2744    <IM:contains rdf:resource="http://www.ict-abovo.eu/ontologies/InformationModel#Gender"/>
2745    <IM:hasAttribute rdf:resource="http://www.ict-abovo.eu/ontologies/InformationModel#ATTa82493e0a9ae4b3d8ea81ac33fe4aec9"/>
2746    <IM:hasAttribute rdf:resource="http://www.ict-abovo.eu/ontologies/InformationModel#ATTc452d2c08f724215856f23e2447a55ef"/>
2747    <owl:sameAs rdf:resource="https://www.upcast-project.eu#TraineeInformation"/>
2748    </rdf:Description>
2749    <rdf:Description rdf:about="http://www.ict-abovo.eu/ontologies/InformationModel#Derive">
2750    <rdf:type rdf:resource="http://www.ict-abovo.eu/ontologies/InformationModel#Operations"/>
2751    <IM:isA rdf:resource="http://www.ict-abovo.eu/ontologies/InformationModel#AnyOperation"/>
2752    <IM:isA rdf:resource="http://www.ict-abovo.eu/ontologies/InformationModel#Obtain"/>
2753    <IM:isAInverse rdf:resource="http://www.ict-abovo.eu/ontologies/InformationModel#Infer"/>
2754    <IM:hasAttribute rdf:resource="http://www.ict-abovo.eu/ontologies/InformationModel#ATT4a0deff147604008ae63885c2463d9df"/>
2755    <IM:inheritedFromInverse rdf:resource="http://www.ict-abovo.eu/ontologies/InformationModel#ATT48aab3b904be4942b5ca33b74b1b0b62"/>
2756    <IM:acceptsHumanActor>false</IM:acceptsHumanActor>
2757    <owl:sameAs rdf:resource="https://w3id.org/dpv/dpv-owl#Derive"/>
2758    <owl:sameAs rdf:resource="http://www.w3.org/ns/odrl/2/derive"/>
2759    </rdf:Description>
2760    <rdf:Description rdf:about="http://www.ict-abovo.eu/ontologies/InformationModel#DOUBLE">
2761    <rdf:type rdf:resource="http://www.ict-abovo.eu/ontologies/InformationModel#AttributeTypes"/>
2762    </rdf:Description>
```

**Figure 6:** Excerpt from the Information Model Ontology

## 3.2. Policy Model alignment

Most goodFlows policy language entities correspond directly to ODRL core vocabulary, reflecting a high degree of conceptual alignment between the two models; Actors map to Assignees, Operations to Actions, and Resources to Targets, while action entities' constraints, contextual constraints and purposes are mapped to ODRL Constraints, ensuring fine-grained access and usage control.

On the other hand, a critical distinction in goodFlows policy model is the explicit handling of pre- and post-actions, i.e., mandatory actions before and after execution, which are mapped to ODRL duties (pre-actions) and obligations (post-actions), respectively. Additionally, goodFlows introduces negative pre- and post-actions, that is, forbidden actions before and after execution, which extend ODRL using prohibitions with temporal constraints.

Table 2 provides the mapping between the goodFlows policy model and ODRL. Leveraging this mapping, goodFlows has been extended with a bidirectional ODRL adaptor that i) converts ODRL rules into the goodFlows policy language, providing for their inclusion into goodFlows reasoning scope, and ii) serialises goodFlows rules to valid ODRL statements.

**Table 2**
goodFlows policy model – ODRL mapping

| goodFlows | ODRL | Comments |
|---|---|---|
| Permission | Permission | Allowing the execution of an action |
| Prohibition | Prohibition | Prohibiting the execution of an action |
| Obligation | Duty | Prescribing the execution of an action |
| Actor Entity | Assignee | The entity performing the action. |
| Operation Entity | Action | The access or usage action performed on a resource (e.g., read, write, delete). |
| Resource Entity | Target | The asset that is being accessed, modified, or controlled. |
| Actor/ Operation/ Resource Entity Constraint | Actor/Action/Target Constraint | Used for the definition of fine-grained constraints upon actors/operations/resources attributes. Both in goodFlows policy language and ODRL, constraints are expressed as tuples of the form <leftOperand, operator, rightOperand> or logical relations thereof. |
| Purpose | Rule Constraint (purpose as leftOperand) | Restricts an action to a specific reason (e.g., research only). |
| Pre-Action | Duty | Action that must occur before the main action is executed. |
| Negative Pre-Action | Prohibition with Constraint (BeforeAction as leftOperand) | Action that must not occur before execution. |
| Post-Action | Obligation | Action that must occur after the main action is executed. |
| Negative Post-Action | Prohibition with Constraint (AfterAction as leftOperand) | Action that must not occur after execution. |
| Contextual Constraint | Rule Constraint | The contextual conditions under which a rule applies. Both in goodFlows policy language and ODRL, contextual conditions are defined as single or logically related constraints at the rule level. |

## 4. Resource consumer-provider conflict resolution

This section describes the process followed by goodFlows to resolve conflicts between the processing intentions of a resource consumer on a third-party dataset, as reflected by a Data Processing Workflow, and the dataset provider's policies regarding the use of the data, specified in ODRL. Figure 7 provides an overview of the process.

The first step is the generation of all intentions that are expressed for the dataset in question through the DPW. To this end, all tasks that consume the dataset are identified, and an Intention object is created based on every execution profile.

An example of an Intention object is shown in Figure 8, corresponding to the task of `PerformStatisticalAnalysis` contained in the workflow that is defined by the data consumer in Figure 5. According to Figure 8, an entity of type `NonGovernmentalOrganisation` is intended to perform statistical analysis on a `TrainingMeasurements` dataset for the purpose of `CommercialResearch`.

The intention objects are passed to the PDP, along with the data provider's policy, specified in ODRL; the associated usage constraints are translated into the goodFlows access and usage control language and imported into a dedicated Policy Model ontology, so as intentions to be evaluated and possibly modified against this ruleset.

For example, Figure 9 presents an ODRL policy that is putting usage constraints on a dataset of type `TrainingMeasurements`; the policy is defined by the data holder for accompanying the dataset when provided to a consumer, and contains two rules:

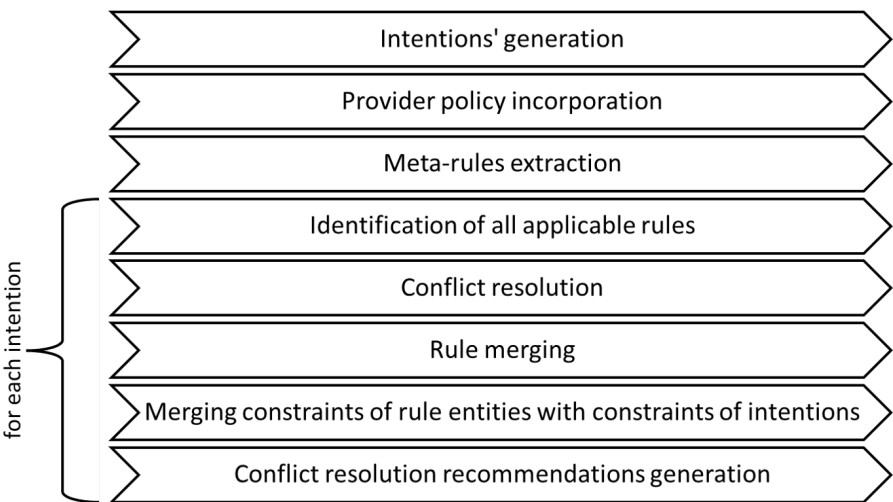

**Figure 7:** Overview of the conflict resolution process

```
 1  {
 2    "intention": {
 3      "purpose": "CommercialResearch",
 4      "actor": {
 5        "@type": "ApiActionEntity",
 6        "concept": "NonGovernmentalOrganisation",
 7        "constraint": null
 8      },
 9      "operation": {
10        "@type": "ApiActionEntity",
11        "concept": "PerformStatisticalAnalysis",
12        "constraint": null
13      },
14      "resource": {
15        "@type": "ApiActionEntity",
16        "concept": "TrainingMeasurements",
17        "constraint": null
18      },
19      "context": null
20    },
21    "usageConstraints": {
22      "@context": [
23        "http://www.w3.org/ns/odrl.jsonld",
24        {
25          "dcat": "http://www.w3.org/ns/dcat#",
26          "dpv": "https://w3id.org/dpv/dpv-owl#",
```

**Figure 8:** Intention object (excerpt)

- A `Permission` (line 14) allowing a `NonGovernmentalOrganisation` with more than 200 employees, for the purpose of `ResearchAndDevelopment`, to partially `Analyse TrainingMeasurements`, by using only `BurntCalories`, `ExerciseType` and `HeartRateRange` contained therein, provided that the associated measurements will be first anonymised (`Anonymise` duty) and that the processing will take place before 31/12/2025.
- A `Prohibition` (line 71) prohibiting a `NonGovernmentalOrganisation` of any size to `Analyse TrainingMeasurements` corresponding to training sessions performed after 1/1/2025 for the purpose of `CommercialResearch`.

In this case the permission is more general applying for the high-level `ResearchAndDevelopment` purpose, while the prohibition constitutes an exception to the permission for the more specific purpose `CommercialResearch` (isA `ResearchAndDevelopment`).

```json
{
  "@context": [
    "http://www.w3.org/ns/odrl.jsonld",
    {
      "dcat": "http://www.w3.org/ns/dcat#",
      "dpv": "https://w3id.org/dpv/dpv-owl#",
      "rdf": "https://www.w3.org/TR/rdf12-schema#",
      "upcast": "https://www.upcast-project.eu#"
    }
  ],
  "@type": "Policy",
  "profile": "http://example.com/odrl:profile:11",
  "uid": "http://example.com/policy:001",
  "permission": [
    {
      "uid": "http://example.com/permission:001",
      "target": [
        {
          "@type": "AssetCollection",
          "source": "https://www.upcast-project.eu#TrainingMeasurements",
          "refinement": [
            {
              "leftOperand": "https://www.upcast-project.eu#Columns",
              "operator": "eq",
              "rightOperand": {
                "@value": "https://www.upcast-project.eu#BurntCalories, https
                    ://www.upcast-project.eu#ExerciseType, https://www.upcast
                    -project.eu#HeartRateRange",
                "@type": "xsd:string"
              }
            }
          ]
        }
      ],
      "assigner": "https://example.com/assigners/Nis",
      "assignee": {
        "@type": "PartyCollection",
        "source": "https://w3id.org/dpv/dpv-owl#NonGovernmentalOrganisation",
        "refinement": [
          {
            "leftOperand": "https://www.upcast-project.eu#NumberOfEmployees",
            "operator": "gteq",
            "rightOperand": {
              "@value": "200",
              "@type": "xsd:integer"
            }
          }
        ]
      },
      "action": "https://w3id.org/dpv/dpv-owl#Analyse",
      "duty": [
        {
          "action": "https://w3id.org/dpv/dpv-owl#Anonymise"
        }
      ],
      "constraint": [
        {
          "leftOperand": "https://w3id.org/dpv/dpv-owl#Purpose",
          "operator": "eq",
          "rightOperandReference": "https://w3id.org/dpv/dpv
              -owl#ResearchAndDevelopment"
        },
        {
          "leftOperand": "dateTime",
          "operator": "lteq",
          "rightOperand": {
            "@value": "2025-12-31",
            "@type": "xsd:date"
          }
        }
      ]
    }
  ],
  "prohibition": [
    {
      "uid": "http://example.com/prohibition:001",
      "target": [
        {
          "@type": "AssetCollection",
          "source": "https://www.upcast-project.eu#TrainingMeasurements",
          "refinement": [
            {
              "leftOperand": "https://www.upcast-project.eu#TrainingDateTime"
              ,
              "operator": "gteq",
              "rightOperand": {
                "@value": "2025-01-01",
                "@type": "xsd:date"
              }
            }
          ]
        }
      ],
      "assigner": "https://example.com/assigners/Nis",
      "assignee": {
        "@type": "PartyCollection",
        "source": "https://w3id.org/dpv/dpv-owl#NonGovernmentalOrganisation"
      },
      "action": "https://w3id.org/dpv/dpv-owl#Analyse",
      "constraint": [
        {
          "leftOperand": "https://w3id.org/dpv/dpv-owl#Purpose",
          "operator": "eq",
          "rightOperandReference": "https://w3id.org/dpv/dpv
              -owl#CommercialResearch"
        }
      ]
    }
  ]
}
```

**Figure 9:** ODRL policy associated with `TrainingMeasurements` dataset

Thereupon, for each rule defined by the provider, the corresponding meta-rules are extracted, as described in section 2.2.

Upon the extraction of the meta-rules, the ground for evaluating intentions against them is set. First, for each intention, the applicable rules, i.e., permissions, prohibitions and obligations affecting the intention, are identified; if no permission is found, the intention cannot be validated (Deny unless permitted).

Assuming at least one permission has been identified (as in Figure 9), conflicting rules are identified, for eliminating overridden rules. This is challenging in the presence of both positive and negative privileges, concept hierarchies and comparison of heterogeneous entities (entities of the action in question, as well as entities associated with pre- and post- actions of rules) with complex constraints that need to be evaluated. The resolution procedure follows several widely adopted patterns ( [24][25][26][27][28]):

- Specific rules override general ones (lex specialis principle): Explicitly defined rules prevail over meta-rules derived by propagation of rules across hierarchies. Moreover, more strict rules (in terms of defined constraints) prevail; for instance, a rule allowing the publishing of training measurements only in EU will prevail over a generic permission for publishing of training measurements without any constraint.
- Deny overrides: if both permissions and prohibitions exist for the same action, prohibitions prevail.
- Inclusion-Exclusion principle for comparing all kind of constraints, pre-actions and contextual conditions; if an action is both permitted and prohibited under different conditions, the system determines whether those conditions overlap.

It is noted that the engine also supports the lex posterior principle, i.e., that most recent rule prevails; however, this pattern is only applicable when an intention is evaluated against internal policies, as in goodFlows the policy author may define a new rule in order to possibly override a previous one or define an exception to a rule.

After the elimination of conflicting rules, the applicable rules are merged, essentially combined into a unified rule that satisfies the intent of all applicable policies, by:

- Merging positive and negative authorisations: a prohibition for executing an action during the night combined with a permission for executing the said action without any temporal conditions set will result to a permission only for daytime.
- Merging constraints and contextual conditions under which results are valid.
- Merging required or forbidden action structures complementing the execution of the requested action.

An additional merge operation that takes place concerns merging the constraints of rule entities with those of intentions, thereby ensuring that constraints defined as part of the intention are taken into account in the compliant intention. This includes comparison of intention and rule constraints, leveraging the Inclusion-Exclusion principle. The lex superior principle applies in this case, i.e., access and usage rules prevail over intentions. It should be noted that in case no valid intention is found, the same procedure is performed for each of the parts of the given resource; possible valid intentions for these parts prescribe projection of the given resource.

This way, an intention may eventually be:

- allowed as-is; however, it may prescribe or forbid the execution of other actions,
- rejected, or
- allowed, but in a modified version, possibly requiring the selection, projection or change of state of resource's fields, along with complementary or forbidden actions.

The latter is the most interesting case, since a recommendation on how to resolve the conflict is generated by the PDP, in the form of an updated intention. For example, the compliant ODRL request for the intention of Figure 8, given the policy of Figure 9, is presented in Figure 10, highlighting the necessary changes to the original request, Both rules defined in the policy of Figure 9 have been applied to the intention of Figure 8, in order to provide common ground between the data consumer and provider, e.g., in the context of a negotiation and eventual agreement. In more detail, the first highlighted modification (lines 61-73) concerns the restriction on the `NumberOfEmployees` property of the assignee, while the second one (lines 102-115) addresses the temporal constraint that the processing can take place before 31/12/2025, both prescribed by the permission of the policy. The third highlighted modification (lines 121-142) concerns refinements of the resource/target of the policy; associated constraints of the original permission and prohibition are combined so that processing is allowed only upon the specific parts (`BurntCalories`, `ExerciseType` and `HeartRateRange`) of the `TrainingMeasurements` and for less recent training sessions performed before 1/1/2025. The fourth modification (lines 9-13) reflects the semantic nature of conflict identification and resolution; although the original rules are defined for the `dpv:Analyse` action, `upcast:PeformStatisticalAnalysis` included in the request does not result in a conflict as `upcast:PeformStatisticalAnalysis` isA `dpv:Analyse` in the Information Model. It is noted that the odrl:implies predicate is leveraged for denoting specialisation of concepts, further fostering explainability. Finally, a duty has been added to the request prescribing anonymisation of the associated measurements before processing (lines 115-119). Figure 11 illustrates the suggestion of possible conflict resolution within the Process Modeller.

## 5. Conclusions

Considering the use case when organisations acquire data from third parties and must align their processing to access and usage constraints set forth by the data providers, this paper has presented three

**Figure 10:** Modified ODRL request

mechanisms towards interoperability and collaboration: an ontology-import mechanism for aligning proprietary concepts with standards such as DPV and ODRL, eliminating manual mapping and enabling rule propagation across class hierarchies; a bidirectional adaptor for translating ODRL to semantic attribute-based access and usage control rules; and an enhanced PDP for the semantic resolution of conflicts among the intentions and constraints of data providers and consumers.

The mechanisms have been built upon the existing goodFlows Policy Engine. This has been driven by two key considerations. First, goodFlows provides a state-of-the-art ABAC implementation, offering a mature and sophisticated reasoning mechanism that aligns well with ODRL's constraint-based policy expression. Although ODRL is not explicitly an ABAC model, its policy constructs and constraints allow it to represent ABAC-like rules effectively. Leveraging goodFlows reasoning capabilities was therefore an intuitive choice, ensuring robust policy evaluation. Furthermore, ODRL's conflict resolution strategy relies on a static conflict property, thereby being insufficient for handling complex rule conflicts in dynamic environments. In real-world scenarios, contextual factors such as attributes of parties and contextual exceptions must influence access control decisions. For instance, while a general rule might

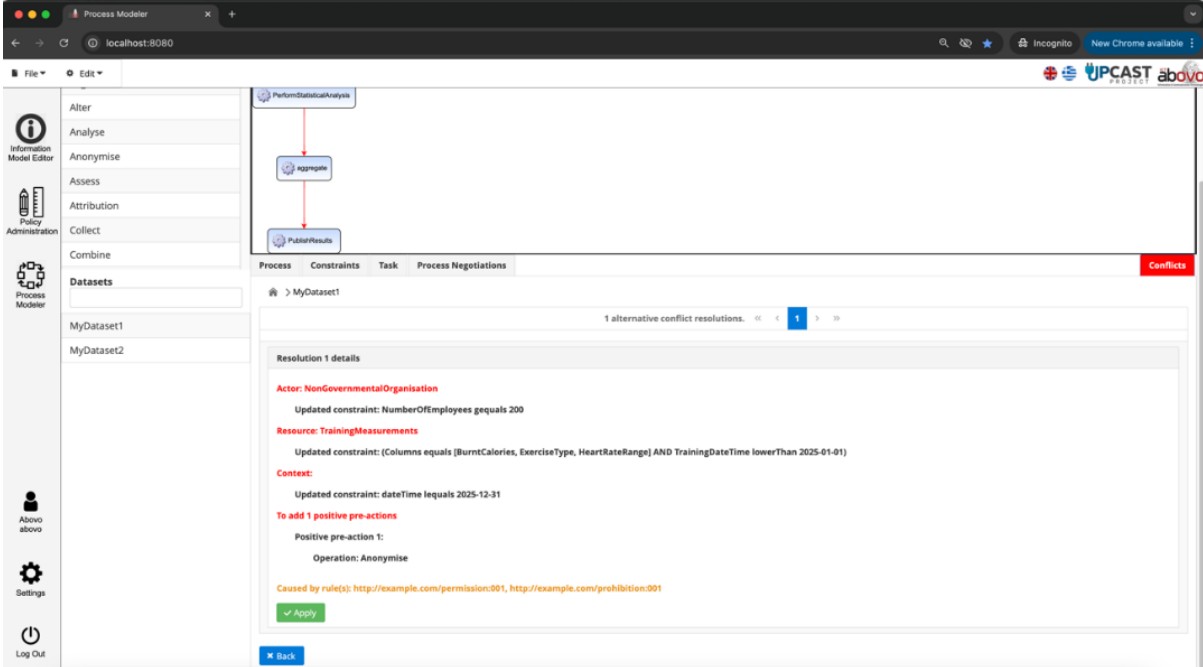

**Figure 11:** Suggestion for conflict resolution in the Process Modeller

prohibit data sharing outside the EU, exceptions based on cross-border agreements or emergency purposes (e.g., a public health crisis) may allow such sharing under specific conditions. Such cases require a more advanced conflict resolution mechanism that can incorporate principles such as lex specialis (specific rules override general ones), lex posterior (newer rules override older ones), and lex superior (higher authority rules override lower ones). Integrating goodFlows' reasoning capabilities with ODRL's policy expression model enables a context-aware, dynamic, and legally grounded access and usage control system that overcomes ODRL's inherent limitations in resolving complex policy conflicts.

## Acknowledgments

This research is being supported by the European Commission, in the frame of the Horizon Europe UPCAST project (Grant No. 101093216).

## Declaration on Generative AI

The authors have not employed any Generative AI tools.

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
