# OpenReview forum: "Semantic conflict resolution for access and usage control"
_SEMANTiCS.cc/2025/Workshop/NXDG — NXDG 2025_

### Official Review · ~Patrick_Hochstenbach1 · 2025-07-14
**Interesting results, good for a demo paper, but it leaves too much open to interpretation by the reader.**

**Rating:** 5
**Confidence:** 4

**Review:**

The contribution Semantic Conflict Resolution for Access and Usage Control describes an extension of the “goodFlows” policy engine to support the acceptance of access and usage policies using standard ontologies such as ODRL and DPV. It also addresses the identification and resolution of conflicts between policies defined by data providers and data processing workflows (DPWs) created by data consumers.

The contribution provides a comprehensive introduction to the goodFlows engine, detailing its internal information, policy, and workflow models from an end-user perspective. It then demonstrates how these models are aligned with standard ontologies such as ODRL and DPV through subclassing. Conflict resolution is presented as a high-level process, illustrated throughout the paper using a running example.

The strength of the paper lies in its demonstration of a potential solution for conflict resolution within the context of the goodFlows engine. However, its main weakness is the limited depth of explanation regarding the inner workings of the goodFlows processing model. The paper offers only a high-level overview, leaving many aspects -- such as constraints on accepted policies, interpretation of the policy semantics, verification of correctness conflict resolution, and completeness (e.g., whether certain types of conflicts may go undetected) -- open to interpretation by the reader.

It is unclear what the precise semantics of the goodFlows Information Model (IM) are, particularly in light of the strong claim that it can be generalized to support “importing and aligning any third-party ontology.” This assertion lacks sufficient justification and raises questions about the model’s flexibility and interoperability. Notably, the use of owl:sameAs to align IM concepts with those from ODRL and DPV is problematic, as it implies full semantic equivalence: an assertion that is both strong and potentially misleading without a detailed semantic mapping or justification.

Section 4 appears to be the core of the paper, as suggested by the title of the contribution. However, the explanation of how conflict resolution actually works -- particularly in the context of the complexities inherent in ODRL policies -- feels somewhat hand-wavy. The discussion is limited to three bullet points on pages 10 and 11, leaving the reader to assume the validity and generalizability of the proposed procedure, especially for use cases that do not rely on goodFlows. If the contribution were intended as a short paper or demo, this level of detail might be acceptable. However, given that it is presented as a full paper and remains well within the 16-page limit (plus additional space for references), a more thorough and concrete explanation and discussion would be both possible and expected to evaluate the validity of the approach.

In general, a considerable amount of space is taken up by screenshots of the Process Modeller, many of which provide more detail than is necessary to understand the paper’s core arguments. The text within these screenshots is often difficult to read, and the figure captions offer little guidance to help interpret what is being shown. As a result, these visuals do not effectively support the reader’s understanding and could be streamlined or better contextualized.

Page 2 : “Figure 1” in paragraph 2 should probably read “Figure 2”.

---

### Official Review · ~Rui_Zhao15 · 2025-07-23
**Review of paper**

**Rating:** 7
**Confidence:** 5

**Review:**

The paper introduces and explains a set of mechanisms to support conflict resolution for an access and usage control policy model in goodFlows, which has a mapping to ODRL and DPV concepts. The rationale is explained well, and introduction is sensible. Through some examples, the paper also explains 1) how goodFlows policy model maps to DPV and ODRL concepts; 2) how ontology importing works (for ODRL and DPV); 3) what do they mean by conflict resolution; 4) how conflict resolution is made (without detailing algorithm / axioms).

I think the paper describes an important topic, and contains interesting contribution to the field, by demonstrating the solution in goodFlows. The results and explanations are sensible and seems sound to me, though formal details are lacking. Overall, it presents a good resource for discussion. On top of that, I have the following comments:

1. There is a lack of explanation of the foundation of the current paper, goodFlows policy model. It is only until at the end there are some examples demonstrating how the different concepts link to each other. The multiple papers cited at that section also brings some confusion for interested readers.

2. It is note made clear what the authors mean by “conflict resolution” in the first half of the paper, leading to uncertainty when reading it.

3. The authors assumed readers have sufficient background in details of ODRL and DPV, e.g. when presenting Sec 3.1. This may be improved.

4. There is a lack of detailed explanation of why the resolution procedure is designed exactly in this way (do the 5 cited papers all use exactly this resolution strategy? if not, what’s the relation between the one in this paper and theirs? are there any internal conflicts in the resolution strategy?)

5. It is not entirely clear how the policies shown in Fig 6, 7 & 8 are related to each other, even after reading last paragraph in Sec 4. In particular, is Fig 6 provided by the data consumer (workflow), and Fig 7 provided by the data providers? Who provides / derives Fig 8? Is it a revised policy for… the data provider, or the data consumer? Is it attached somewhere, or just internally used by the reasoner / policy engine?


In addition, in the background section, the authors did explain how this work relates to work like ODRL. But it would benefit if the authors also discuss the difference between their work with respect to other policy work taking a similar view, that data providers and data consumers have different policies which interact. For example, Fig 1 reminded me of Fig 2 in

- Dr.Aid: Supporting Data-Governance Rule Compliance for Decentralized Collaboration in an Automated Way. CSCW 2021

And therefore some other related papers come to my mind, e.g.:

- Perennial Semantic Data Terms of Use for Decentralized Web. *WWW 2024*

- Decentralizing Privacy Enforcement for Internet of Things Smart Objects. *Computer Networks*, 2018

---

### Decision · Program_Chairs · 2025-07-25

Accept